# Discriminator Based Corpus Generation for General Code Synthesis

## Abstract

Current work on neural code synthesis consists of increasingly sophisticated architectures being trained on highly simplified domain-specific languages, using uniform sampling across program space of those languages for training. By comparison, program space for a C-like language is vast, and extremely sparsely populated in terms of 'useful' functionalities; this requires a far more intelligent approach to corpus generation for effective training. We use a genetic programming approach using an iteratively retrained *discriminator* to produce a population suitable as labelled training data for a neural code synthesis architecture. We demonstrate that use of a discriminator-based training corpus generator, trained using only unlabelled problem specifications in classic Programming-by-Example format, greatly improves network performance compared to current uniform sampling techniques.

## 1 Introduction

Automated code synthesis is increasingly being studied as a way to lower the entry bar for non-experts to create computer software, and to aid in generally taming the complexity of large-scale systems by allowing engineers to specify their intentions at a higher level of abstraction. The approach of neural code synthesis in particular has recently gained a lot of attention, applying advances in neural networks to the problem of automated synthesis.

We specifically study the approach of *programming by example*, in which a small set of input-output examples are presented to the system to serve as a guide to the desired functionality of a program. Based on an analysis of these examples the synthesis system returns a source-code program able to replicate that functionality. Recent research in this field demonstrates promising results, including DeepCoder Balog et al. (2017) and Zohar & Wolf (2018). However, research to date is limited to using domain-specific languages and often linear sequential programs without conditions or loops.

We also take a neural-network-based approach to this problem in an attempt to gain inter-program inference across the training examples given to our system, potentially allowing the system to learn general aspects of programming to help synthesize new programs from unseen input/output examples. Unlike existing recent work, however, we target a general-purpose low-level programming language for code synthesis with a much larger search space of possible programs. This presents a major challenge in generating a *training corpus* for the neural network. Where related research has used uniform sampling methods through program search space (Sun et al. (2018)Chen et al. (2017)), or even enumerative approaches(Balog et al. (2017)), such approaches are wholly inadequate over larger search volumes – with sparse sampling producing very poor inference results.

To solve this training corpus generation problem we propose a novel discriminator-based system, in which new sub-corpora are iteratively created, continually measuring their functional properties against those of the problems it is attempting to solve. This process works by learning how similar the I/O mappings of generated programs are to I/O problems requested by users; by selecting programs which result in increasingly similar I/O mappings we simultaneously choose programs with similar underlying source code features, until we are able to solve I/O problems requested by users. We demonstrate that the resultant training corpus is greatly superior to a conventionally generated corpus via uniform sampling, when using a more generalised programming language for synthesis.

We measure the performance of our approach by comparing against similar research on neural code synthesis which uses uniform or enumerative sampling for training, demonstrating that our

discriminator-informed corpus generation approach far exceeds uniform sampling, by a factor of 2, in terms of find-rates. We also compare against a general baseline using genetic programming (GP); this baseline produces a surprising result that GP has a broader range of programs found, although its probability of resolving any given user-provided problem is worse.

Our approach offers an effective way to generate a training corpus for a high-dimensional program search space, capable of finding a wide range of unseen useful programs based only on input/output examples, without any labelled training data. At a high level our research also demonstrates that the structure of the training corpus provided to a neural network greatly affects its performance on general purpose code generation tasks, and we argue that it should therefore represent a core focus of the code synthesis community's efforts alongside work on neural network and language structures.

In the remainder of this paper we firstly assess the literature in the field, focusing on neural code synthesis and specifically its corpus generation techniques. In Sec. 3 we then present the methodology we use to build our system, based on both a synthesis network and a discriminator network for corpus generation. We then evaluate our approach in Sec. 4 by comparing it against traditional training corpus generation approaches for neural code synthesis.

**[code to reproduce our results will be made open-source should this paper be accepted, and this line will be changed to the link to the repository]**

## 2 BACKGROUND

Program synthesis has been studied for nearly as long as machine learning has been considered, having been proposed in 1950 by Alan Turing Turing (1950).

Three main subfields exist: logic-based solvers Feng et al. (2017; 2018); Feser et al. (2015a;b); Solar-Lezama et al. (2006); stochastic-search-based Genetic Programming (GP) Koza (2010); Haraldsson & Woodward (2014); and neural network based approaches Kant (2018). While solvers are excellent for restricted domains, they have limited applicability in programs containing loops due to their reliance on predicate logic. Genetic programming has been used to good effect in specific domains, but lacks the cross-program inference power that neural networks offer. We hold neural code synthesis to be an interesting area of study following the results of DeepCoder (Balog et al. (2017)), which suggests that neural network can identify higher level features of program outputs (sortedness, evenness, offset from zero mean...); combined with this is the opportunity of inter-program inference within the representational model of the neural network.

Within the subfield of neural synthesis, we focus on programming by example Gulwani & Jain (2017), which takes a set of inputs and outputs which demonstrate a desired functionality.

One of the core works in this field is DeepCoder Balog et al. (2017), a system able to recognise sub-functions within a larger program consisting of up to 5 of these sub-functions called sequentially. This was significantly improved in subsequent work Zohar & Wolf (2018), which used a novel partial program execution model with results fed back into the synthesis neural network, to enhance search speed and increase the length of programs that can be found up to 14. Both approaches operated in the same Domain Specific Language, which focused on arrays of integers. In terms of corpus generation, DeepCoder uses a simple enumerative strategy, as its DSL creates a small enough program space to list all possible programs, while Zohar & Wolf (2018) use random sampling.

Two other major domains exist in the field of neural code synthesis; In both of these, we see that synthetic training sets are predominantly generated by uniform sampling ( Sun et al. (2018); Chen et al. (2017); Devlin et al. (2017)).

The first of these two domains is string-manipulation functions Gulwani (2011); Raza & Gulwani (2017), as these have immediate and obvious usefulness to non-expert users. While these tasks can be handled by solvers, there are good applications for neural synthesis systems, such as RobustFill Devlin et al. (2017), or neurally-augmented solver systems Vijayakumar et al. (2018).

The second domain is deduction of the behavioural algorithm of an agent in an environment, with the examples being sequences of actions taken by the agent in a grid environment. This has been studied both in terms of trained neural networks recognising behavioural patterns Sun et al. (2018)Bunel et al. (2018) or in reinforcement learning based approaches to program generation Chen et al. (2017).

Our work differs from these domains' use of random uniform sampling for training sets, as well as from the unscalable enumerative strategy of DeepCoder, in that we introduce a strategy for generating a targeted training corpus without requirement for human-provided labelled data, and we investigate how this generation strategy significantly boosts inference performance.

Finally, the most similar work to our own is Shin et al. (2019), which investigate the effects of *inputs to programs* in terms of the produced I/O examples' effect on training success. This differs from our work in that they explore which inputs to feed into sampled programs, while we are presenting a new program sampling technique. This approach may well be complementary to our own if using as part of our training corpus generation pipeline.

## 3 METHODOLOGY

Our overall approach to code synthesis is to feed in a set of ten input/output (I/O) examples to a neural network, and have the output layers of that neural network select the most likely operation for each line of code for a function of a given upper length limit (where the number of output layers is equivalent to the maximum number of lines of code in the function, each neuron in an output layer is equivalent to selecting one particular operator for that line, and each line of code can be set to 'no-op'). The probabilistic nature of a neural network's output allows us to gain a set of possible programs to search through up to a given search depth; this basic approach is similar in spirit to DeepCoder Balog et al. (2017) in that it is attempting to guess the probability of each 'feature' of a program – except in our case these features are relatively low-level instructions.

Neural networks for code synthesis are usually trained on a uniformly sampled set of programs from the total space of all possible programs; while this is viable for simple domain-specific languages, it becomes intractable for more general purpose languages. With the aim of achieving code synthesis using a much more general programming language, our methodology focuses on how training data can be effectively drawn from a much larger search space without human input.

Our approach to this uses a novel hybrid solution inspired by genetic programming but using a neural network as a fitness function. We term this neural network a *discriminator*, as it attempts to discriminate between the algorithms the genetic programming element is currently generating, and the likely features needed by requested I/O examples for human-useful functions.

### 3.1 LANGUAGE USED

We reuse the programming language used in Wild & Porter (2019), a simplified C-like language, which can be compiled out to C or Java source code. The language features variable assignments, conditional operators, and loop operators. These allow complex flow-control operations, allowing us to study algorithms which go beyond linear concatenation of functions often used in the literature.

We enhance the problem complexity considered by Wild & Porter (2019) by removing the restrictions on array length, instead implementing an operation which instantiates empty arrays of given length. We also implement operations to allow array length to be determined, and the ability for a function to call itself, to allow recursive functions. Our methodology also avoids using any human-provided hints about the likely features of source code for problem solutions, instead relying only on a set of I/O examples for unsolved problems.

The full set of language operators available and restrictions used is included as an appendix 6.1.

### 3.2 HUMAN USEFUL CORPUS

Our human useful corpus is a set of I/O mappings for 40 unique functions, each of which takes one array input parameter (of any length) and one variable, and returns an array of any length. The set of problems includes reversing arrays, appending arrays with new values, and summing the values in an array (a full list is given in appendix 6.4). We assume that these I/O mappings have been requested by human users, but that none have yet been solved. We further assume that each function has been requested at least five times, with different I/O mappings for each. This gives a total corpus of 200 I/O examples as a guide to the kinds of input-to-output transformations that are considered 'useful'. We stress that at no point is the system ever provided with any source code for these examples.

During early experimentation we found it beneficial to apply some conventions around these I/O examples, specifically for the first three examples in the set of ten for a given problem. The first I/O example for any problem is such that the content of each input array cell is the index of that cell, starting from 0, and the input variable has a random value. The second I/O example has the same properties but the second array cell is randomised to a new value between 8 and 8, inclusive. The third I/O example for every problem is the same as the first example except that the input variable is also randomised to the same range. The remaining seven I/O examples can be anything, and are randomly generated in our corpus.

Our synthesis pipeline has the challenge of starting from this corpus of unsolved problems, specified by I/O examples, and solving as many of them as possible by synthesising the correct source code which correctly maps the given input to the given output for each problem.

### 3.3 NEURAL NETWORK ARCHITECTURE AND SEARCH PROCESS

Two neural network architectures are used in our synthesis pipeline, a synthesiser and a discriminator. The discriminator is discussed later, and is used in generating corpora on which the synthesiser is trained. The synthesiser network is that which receives I/O examples and attempts to build a source code program to match the required functionality.

This neural network has an input layer which accepts both the input and output values of each pair of 10 examples for a given problem, such that each input neuron takes a single bit of each integer. Internally we use of a set of 8 layers of 256 neurons, each connected to all previous layers, with selu activation, a simplified version of the net used in Zohar & Wolf (2018). For output, we have one output layer per line of the program to synthesise. Each output layer has one neuron for each way in which a line can be written (all valid operations), including no-op. A labelled program would be represented as an array of one-hot vectors, with the non-zero value mapping to the way that particular line should be written. We use crossentropy training loss on each line.

When reading a program out, the activities of all output layers' neurons are taken and ranked, giving a confidence for each option for each line. We then search over the top 1,000,000 programs the network returns as 'most confident', using a beam search technique (detailed in appendix 6.2).

### 3.4 INITIAL SUB-CORPUS GENERATION

Our synthesiser neural network requires a training corpus comprised of the source code of example programs together with the I/O pairs for each program. Based on this training it is then able to solve some of the problems in our set of human useful I/O examples.

To generate this training corpus we use an iterative process of genetic programming and discriminator training to create a series of increasingly relevant corpora. At the start of this process, we generate an initial corpus of 1,000 functionally unique programs sampled at random from the total space of all possible programs (where we quantify a program as unique if at least one output value is different from any other program when given the same five randomly-generated input parameters).

Corpora other than this starting corpus are created by using a parent corpus from the set of accepted corpora (creation process detailed below). A child corpus is accepted if it finds an implementation for a human-useful IO mapping which was not found by an existing accepted lower-ancestry corpus, otherwise it is discarded and cannot be used as a parent. A corpus' ancestry is the number of parent-child relationships it is removed from the initial corpus.

When creating a new child corpus, a parent corpus is selected by roulette selection from all currently accepted potential parents, with each potential parent's weighting being $(0.1 + number\_of\_successful\_children)/(0.1 + number\_of\_children)$.

### 3.5 DISCRIMINATOR TRAINING AND USAGE IN SUB-CORPUS INHERITANCE

After the first corpus, further corpora are generated based on the use of our discriminator. This is a neural network designed to classify input/output pairs generated by programs in our generated corpus as being *closer to / further away from*, those of I/O pairs in the human-useful set from users.

A new corpus is created by selecting programs from a parent corpus which are measured to be most similar to human-useful programs in their input/output mappings (specifically the form of

their outputs, and how these outputs seem to relate to corresponding inputs). By doing this, we hypothesise that the kinds of source code features found in these programs will similarly move closer to those needed to synthesise programs solving the human-useful I/O examples.

Our discriminator neural network then works as follows. Architecturally, it consists of 2 dense layers of 16 nodes, with a single output. The featurisation of programs is identical to our synthesis neural network (as described above). We train this network by providing all of our I/O examples for all unfound human-useful target programs, and generate I/O examples for each of the 1,000 programs in the parent corpus. We train the discriminator network as a classifier to determine which of the I/O examples are from our human-useful set, and which are from the generated set. This training continues until a threshold $T_k$ is reached. $T_k$ is the proportion of programs which would be retained by the discriminator (as described below), if run on the parent corpus. $T_k$ is a randomly set value between 0.1 and 1, set as $max(0.1, r^2)$ where $r$ is a random real value uniformly distributed between 0 and 1. We use a random value here to increase the diversity in corpora formed, some being highly similar to the parent and some being fairly different, in a bid to maximise coverage.

This trained discriminator therefore returns an estimate $F_d$ for how *human-like* a program is, ranging between 0 and 1. A program is said to pass the discriminator if it has an estimate of $F_d > 0.1$. This second threshold was chosen based on preliminary experiments, particularly based on analysis of the distribution of estimates, which was found to be highly biased towards either end of the spectrum.

These selected programs form the basis of a new child corpus. This child corpus is then expanded to have 1,000 functionally unique programs of its own, by using roulette wheel selection in which we take one of the existing programs in the corpus and mutate it, then accept or reject that mutated program as an additional member of the corpus based on a fitness function $F_q$. This value $F_q$ is simply how much it exceeded the discrimination threshold ($F_d - 0.1$). During development, roulette selection was found to produce far superior results than tournament selection if the discriminator values are offset by $0.1$, due to bias away from programs which only just passed the threshold.

We continue to create new child corpora in this way up to a desired total number. We emphasise that, during this process, we have no access to the human-useful programs, only the I/O examples that users have requested be generated. After multiple rounds of generating self-training data in the above fashion, to reach source code features that are increasingly likely to be involved in solving the requested I/O examples, we then begin to be able to successfully synthesise solutions to the I/O examples of programs requested by users.

### 3.6 EVALUATION OF OVERALL SYSTEM

Our synthesis pipeline returns a collection of solutions to I/O problems in the human useful set, and also returns a set of generated training corpora which were used in finding these solutions. We keep all generated corpora, and their corresponding trained synthesiser networks, which we 'kept' in the above iterative process as either a parent or final child. Note that we keep the intermediate parents along the way because a child corpus is sometimes unable to solve some of the problems of its parent, even though it can solve new problems that its parent could not.

These trained neural networks can then be re-used when given new I/O problems not present in the initial human useful set; each individual trained synthesiser network processes an I/O problem in $\sim 0.25$ seconds. We can also collate all corpora into a single combined corpus on which to train a combined synthesiser neural network. We report the results of both alternatives (the set of individual experts and the combined expert) in the following section.

## 4 RESULTS

Our evaluation compares our approach to training corpus generation against competitive baselines. For all experiments we generate a total of 200 sub-corpora using our approach, with each experiment repeated 20 times. The HU corpus was fixed for all experiments, and the I/O examples did not vary, including in baselines, to allow consistent testing and repeatability. Noise between repeated experiments therefore derives only from the different systems' internal stochasticity.

To evaluate our system we compare it against two baselines: one without using our discriminator, instead using randomly-generated corpora, and one using genetic programming on the same problem set. We then compare our approach to one using uniformly-sampled training data, as is used in

Table 1: Find rates for our human-useful problem set by approach type. The set consists of 200 I/O examples, derived from 40 ground-truth programs

| Approach | I/O examples solved | Unique functionalities found |
|---|---|---|
| Discriminated sub-corpora | $\mathbf{81.5}(\sigma = \mathbf{8.28})$ | $20.3(\sigma = 2.08)$ |
| Random sub-corpora | $43.5(\sigma = 4.29)$ | $11.8(\sigma = 1.20)$ |
| Genetic Programming | $64.6(\sigma = 3.37)$ | $\mathbf{22.5}(\sigma = \mathbf{1.86})$ |

related research for simpler domain-specific languages, and lastly we explore the the effects of decisions made by our discriminator in more detail.

## 4.1 PERFORMANCE COMPARED TO BASELINES

In this experiment we compare the find rates of programs against two baselines. For the first baseline we use an identical system to our own but with the discriminator removed as a fitness function, such that successive training corpora are generated only using randomly selected parents and mutations. The remainder of our pipeline is kept the same for this baseline.

For our second baseline, we use genetic programming due to the inability to re-use baselines from other work in the literature, as for example seen in the DeepCoder framework, caused by the relative generality of our programming language for synthesis. Our genetic programming technique was designed to require roughly the same total computational time as the full sub-corpus generation pass, to fairly compare the options for user IO mapping resolution. It uses a population size of 1,000, and a maximum generation count of 10,000. It uses tournament selection, with a tournament size of 6 and a probability of mutation of 0.15. The fitness function is a function of the Euclidean distance between the desired output and the target output, unless the outputs differ in length, in which case it is set to an extremely negative penalty value.

Our results are shown in Table 1, demonstrating that the set of networks produced by discriminated sub-corpus generation between them produced the highest resolution rate for the 200 human-useful I/O mappings, returning an average of $81.5(\sigma = 8.3)$.

Interestingly, the genetic programming technique found more unique functionalities. Of the 40 unique ground truth programs, the GP baseline found 22.7, while the sub-corpora found 20.3. This means that while the sub-corpus technique had a higher probability of finding a program to match a user-supplied IO mapping, there existed a stronger inequality between the find rates than the GP algorithm, which was more consistent in its probability of finding any given program.

It is highly likely that this is a phenomenon produced by the nature of the discriminator itself, driving the system towards producing certain types of algorithms predominantly. It seems likely that certain properties of certain programs would be more recognisable by a neural network, and so emphasised by the discriminator's fitness function; this is a key avenue of future work.

## 4.2 COMPARISON WITH UNIFORMLY GENERATED TRAINING DATA

In this set of experiments we evaluate the performance of our iteratively-produced training corpus against a baseline randomly-generated training corpus. To do this we take a subset of 5 of the sub-corpora produced, attempting to maximise the number of separate I/O examples found by the set of networks. The collated corpus then discards any duplicated functionalities, giving a training corpus of approximately 4,200 examples. For each of these, we write out a set of training examples, with 5 randomly generated inputs into each function being used to generate the feature I/O examples. We then train the same synthesis neural network on these as before.

For the comparative baseline training data we simply sample 5,000 programs uniformly at random from the space of all possible programs, and again generate 5 training I/O examples using each of these programs. In both cases the corpora are then divided between training and validation in a 0.9:0.1 split, with validation programs being functionally distinct from those in the training set; the training data is then fed into our synthesis neural network and tested.

As can be seen in Table 2, the collated corpus produced by the discriminated sub-corpus generation process more than doubles the performance of the randomly generated training corpus. The discrim-

Table 2:

| Approach | I/O examples solved | Unique functionalities found |
|---|---|---|
| Collated discriminated corpus | $\mathbf{44.5}(\sigma = \mathbf{7.5})$ | $\mathbf{11.3}(\sigma = \mathbf{2})$ |
| Random corpus | $16.1(\sigma = 2.5)$ | $3.85(\sigma = 0.59)$ |

inator has driven program generation towards a set of programs which is far more representative of the types of behaviour present within the human-useful programs.

Inspection of the programs within the corpora matches this expectation: there is no drive for a randomly generated program to, say, contain a loop, and if they do there is no drive towards writing to all/most cells in the output array. While the behaviour of each training program is thus distinct, this is not necessarily in a 'useful' way, relative to the types of problems the user wishes to solve.

The discriminator, however, can drive towards useful training programs, by emphasising programs which write relatively uniformly to output arrays, using loops and with values remaining within sensible ranges; as well as generating programs with outputs dependent on the inputs, rather than fixed-output programs. Without ever receiving training labels or information about the nature of useful features, the discriminator learns to maximise their presence in the sampled programs.

### 4.3 ANALYSIS OF EFFECTS OF ITERATED DISCRIMINATOR USE

In this section we explore in more detail of effect of our discriminator design. We first demonstrate the difference-over-time from our first baseline in Sec. 4.1, in which we generate successive corpora using either our discriminator or using simple random selection of parent. The results of this are shown in Figure 1, in which we see how many programs from our target human useful set are solved over time as successive corpora are generated. As expected, both the discriminator setup and the random setup start at the same point, as the initial corpus does not use a discriminator. The first discriminated sub-corpus trains a network which finds an additional 6.7 solutions to I/O examples on average, compared to the non-discriminated corpus' increase of only 2.8. This progress continues as new sub-corpora are added, and is also seen on the graph of unique functionalities found. This clearly demonstrates the value of the discriminator in corpus generation.

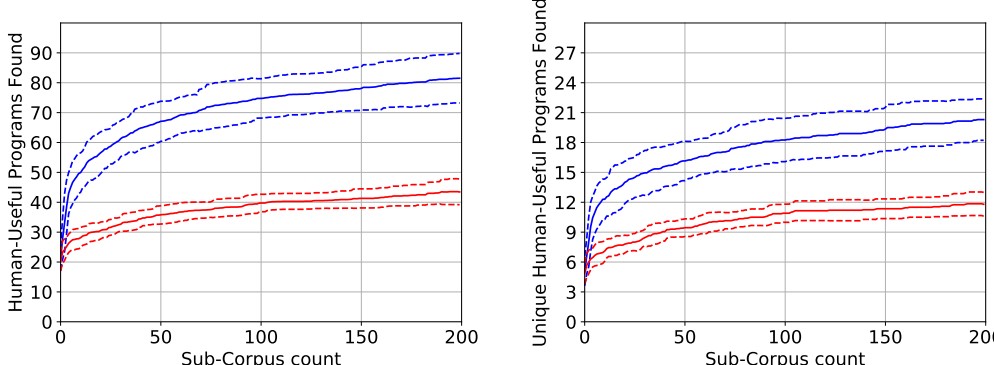

Figure 1: Find rates of programs satisfying IO specifications, over sub-corpus count, for both our approach (blue, higher) and random baseline (red, lower). Find rates for all 200 I/O specifications on left, find rates for unique functionalities on right. First standard deviation shown.

We next examine the find rates over the 'ancestry' of sub-corpora in detail, to give indications as to the behaviour of the system in response to the discriminator's iterated use. Each sub-corpus past the first uses another as a parent; the ancestry of a sub-corpus is therefore the number of parents since the starting corpus. This varied by experiment, with all accepting a corpus with ancestry of at least 5, and the maximum being a single run which had a sub-corpus of ancestry 12.

In Figure 2 we plot the find rates of each program over ancestry progression. We discover that certain programs are trivial to find, and do not require discriminator use (Array Length; Array to Zero...). These have a find rate of nearly 1.0 before the discriminator is used (black in first cell).

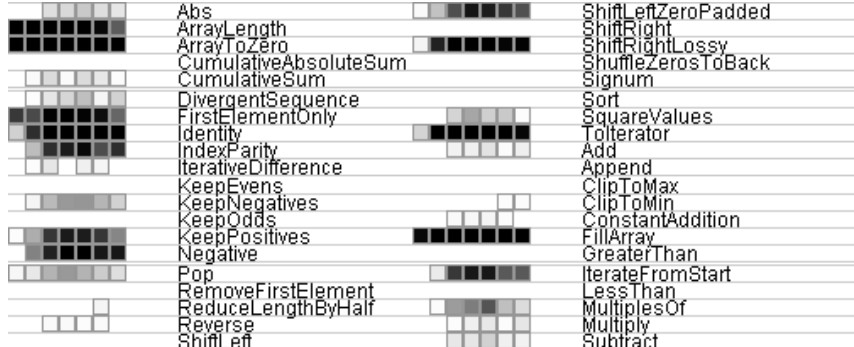

Figure 2: Program find rates over sub-corpus ancestry. From white at 0 find rate, to fully black indicating 1.0 find rate. Border used to indicate a non-zero value. Plot cut off at $ancestry = 8$ due to low sample size past this point. The way actual program outputs evolve is detailed in appendix 6.3.

Certain programs are found with high reliability past ancestry of 1, for example the identity program. This program was almost never found without use of the discriminator, but a single use lead to it having a nearly 1.0 find rate. This indicates that the discriminator lead to a set of sub-corpora which represented the identity function's programmatic behaviour far better. We found that these sub-corpora's programs nearly universally featured loops and sequential array write operations, functionalities required to produce the identity function.

The second use of the discriminator showed similar programs, such as Identity Parity and 'Iterate from Start'. These were rarely found in both a non-discriminated sub-corpus, or an $ancestry = 1$ sub-corpus, but featured regularly at later depths. This reflects the discriminator iterating on its previous selections, attempting to discriminate between programs produced by a first-generation discriminator and the human useful corpus. These programs now found feature more complex loop-using behaviours than simple reproduction of the input array, such as using conditionals and literals.

Past $ancestry = 2$, however, we no longer see sudden find-rate jumps. This indicates that the discriminator loses its effectiveness and can no longer guide as reliably towards the functionalities of the human-useful corpus. We speculate that the discriminator is unable to force the presence of the required functionalities in the produced training corpus either because (i) the genetic algorithm doesn't produce any for it to select; (ii) it does not have the capacity to represent the behaviours (due to low layer width or depth); (iii) these functionalities are simply not identifiable from I/O mappings alone. Further study of this is the subject of our future work.

## 5    CONCLUSION

This paper has presented a discriminator-based corpus generation technique, which iteratively seeks to generate training programs drawn from the same distribution as the programs it is attempting to solve. It works without the need for labelled training data, generating its own based purely on supplied features of I/O examples and the underlying properties of the language itself. We show that it is greatly superior to one which does not use a discriminator for selecting training examples.

Once generation has completed, our framework can also return a collated training corpus, allowing training of a single large neural network. We show that this collated network is also significantly stronger, in terms of quality of trained network, to one trained using random sampling techniques.

Based on our results, we argue that the way in which training corpora are generated for neural program synthesis deserves significant further study – and may be of equal importance to the design of the neural network used for synthesis itself. In future work we will further explore the ability of discriminator-style networks to identify specific features of code likely to be involved in solving a particular problem, as well as more advanced architectures for synthesis and discriminator networks.

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

# 6 APPENDICES

## 6.1 LANGUAGE IMPLEMENTATION DETAILS

Our programs are implemented as a constrained version of C, as in the work of Wild & Porter (2019). We, however, expand upon their work, increasing the limits on variable count, and introducing the concepts of runtime-defined array lengths and recursive function calls. We limit the programs to length 14, the maximum number of integer variables to 7, the maximum number of array variables to 2 (the one input and the one ultimately returned), the maximum array length under any condition to 16 and the maximum value ever to between -255 and 255 (inclusive).

The operators available are described in Table 3, and represent a superset of the operators used in the work of Wild and Porter.

Table 3: The 16 operators available in implementation of language, with the inputs to the functions and their purpose

| Operator | Input |
| --- | --- |
| Assign from Variable to Array | Array: Write Target; Integer: Array Index; Integer: Read Target |
| Assign from Variable to Array | Array: Read Target; Integer: Array Index; Integer: Write Target |
| Create Array | Array: Array to initialise/overwrite; Integer: Length of Array |
| Get Array Length | Array: Array to read length of; Integer: Write Target |
| Variable to Literal | Integer: Write Target; Literal: Can be -1, 0 or 1 |
| Arithmetic Add | Integer: Write Target; Integer: Read 1; Integer: Read 2 |
| Arithmetic Subtract | Integer: Write Target; Integer: Read 1; Integer: Read 2 |
| Arithmetic Multiply | Integer: Write Target; Integer: Read 1; Integer: Read 2 |
| Arithmetic Divide | Integer: Write Target; Integer: Read 1; Integer: Read 2 |
| Arithmetic Modulo | Integer: Write Target; Integer: Read 1; Integer: Read 2 |
| Assign Variable to other Variable | Integer: Write Target; Integer: Read 1 |
| Flow: Loop | Integer: Iterator; Integer: Loop to |
| Flow: Condition: Greater than 0 | Integer: Variable to test |
| Flow: Condition: Integer equality | Integer: Variable to test 1; Integer: Variable to test 2 |
| Recursive Function Call | Array: Array to set to function return; Array: Input 1; Integer: Input 2 |
| No-Operation | No Inputs |

## 6.2 Beam Search

Each line is given a "depth" to search, ranging from a single option to up to a maximum of 10 options. The combination of all programs within this space are initially added to a set, which therefore has a size equal to the product of all the depths. The beam search constructs the initial search volume by iteratively increasing the search depth of a single line to maximise the exploration value function. This function is as follows:

$[a_0, a_1, a_2...a_n]$ represents the ranked option confidences from the neural network, now sorted such that $a_0$ is the highest-confidence option. These are normalised by dividing by $a_0$.

$S$ is the search volume size, and $S_i'$ is the search volume size if line $i$ had its depth increased by one.

$D_i$ is the current depth of search of a given line minus one (for example if $D_i$ is 0, only the first option will be added to the search volume, if it is 1, the top two ranked options will be added to the combinational set of programs (thus doubling the search space size)).

The exploration value for adding increasing the depth of any given line $i$ is $\frac{(a_{D_i+1})*0.75^i}{S_i'-S}$

This process attempts to drive the neural network towards exploring lines in which multiple options are highly confident, and away from lines where a single option has been given a high confidence and all others given a low or negative confidence.

Once the iterative addition has produced a set of $>= 1,000,000$, the 1,000,000 options with the highest sum confidence are selected and searched exhaustively.

## 6.3 Generated program examples

In Table 4 we see outputs from 15 randomly generated programs, from 3 randomly chosen sub-corpora. The first is the starting corpus of the run, which had no discriminator. The second has a discriminator trained between the human-useful IO examples and its parent corpus. The third sub-corpus then has a second generation discriminator, which was trained based on a discriminated corpus and the HU IO corpus.

All outputs are responses to the function being run with an input of input_array = [0,1,2,3,4,5,6,7] and input_integer = 2.

We see that the programs in the starting corpus, ancestry=0, which were randomly sampled from program space, differ greatly from the style of program we are attempting to train the network to synthesise. The majority of all returned values are 0, and the array length varies considerably. There is little evidence that the input array is being read in, or indeed any use of loops at all.

The second generation corpus, ancestry=1, shows little use of the input values, but has outputs in more consistent ranges. The output lengths now appear to always be the length of the input array, and the programs clearly use loops to write to the output array. Despite this, the output patterns are highly uniform, often being the same value repeated for most of the output array.

The third generation corpus, ancestry=2, shows more complex program still. Negative values, which would require arithmetic operations to produce, are present. The values vary across larger ranges, and they show elements of the input array (the last example being the input array changed by a single element).

We believe this is fairly illustrative of the behaviour of the discriminator, although more in-depth analysis of the programs produced could form a good direction for future work.

## 6.4 Corpus of target 'Human Useful' functions

Functionally, all programs within this corpus receive both an array of length 2 to 16, with values ranging between -8 and 8, and an integer also ranging from -8 to 8, the same as in the rest of the paper. Some functions simply do not use the input variable, but this is accomplished solely by ignoring it, not by changing the function input parameter set.

The human useful programs and their functionalities are listed in table 5

Table 4: Examples of outputs of generated programs, randomly chosen from 3 sub-corpora. These sub-corpora are organised based on ancestry, how many parent-child relationships exist between the first sub-corpus (ancestry=0).

| Sub-corpus ancestry | Program Output |
| --- | --- |
| Ancestry=0 | [0] |
| Ancestry=0 | [0,0,0,0,0,0,0,0] |
| Ancestry=0 | [0,0,0,0,0,8,0,0] |
| Ancestry=0 | [2] |
| Ancestry=0 | [0,0,0,0,0,0,0,0,0,0,0,0,0,0,0,0,0] |
| | |
| Ancestry=1 | [7,7,7,8,7,7,7,7] |
| Ancestry=1 | [0,2,2,2,2,2,2,0] |
| Ancestry=1 | [7,8,8,8,8,8,8,8] |
| Ancestry=1 | [8,8,8,2,8,8,8,8] |
| Ancestry=1 | [2,2,2,2,2,2,0,0] |
| | |
| Ancestry=2 | [0,8,16,24,32,40,48,56] |
| Ancestry=2 | [-1,-7,-6,-5,-4,-3,-2,-1] |
| Ancestry=2 | [0,8,16,24,32,40,48,56] |
| Ancestry=2 | [0,-7,-7,-7,-7,-7,-7,-7] |
| Ancestry=2 | [-2,1,2,3,4,5,6,7] |

Table 5: Human useful programs, specified by user and presented to the system as a set of IO mappings only.

| Program | Function |
| --- | --- |
| Absolute Values | Returns an array of equal size to the input array, where all values are the absolute values in the input array |
| Array Length | Returns a 1-length array, whose value is the length of the input array |
| Array to Zero | Returns an array of equal length to the input array, filled with zeroes |
| Cumulative Absolute Sum | Returns an array of equal length to the input array, filled with the cumulative absolute values. That is to say cell $output_i = \sum_{n=0}^{i} |input_n|$ |
| Cumulative Sum | Returns an array of equal length to the input array, filled with the cumulative values. That is to say cell $output_i = \sum_{n=0}^{i} input_n$ |
| Divergent Sequence | Returns an array of equal length to the input array, filled with values such that if $i$ is even $output_i = i/2$, otherwise $output_i = -(i/2)$, e.g. [0,0,1,-1,2,-2...] |
| First Element Only | Returns an array of length 1, whose content is $input_0$ |
| Identity | Returns a new array with length and contents matching the input array |
| Index Parity | Returns a new array with length equal to that of the input array, filled with alternating 0s and 1s, e.g. [0,1,0,1...] |
| Iterative Difference | Returns a new array with length equal to that of the input array, filled with the difference between the mapped input array cell and the next, such that $output_i = input_i - input_{i-1}$, with $output_0 = input_0$ |
| Keep Evens | Returns an array of equal length to that of the input array. If $input_i$ is even, $output_i = input_i$ else $output_i = 0$ |
| Keep Negatives | Returns an array of equal length to that of the input array. If $input_i < 0$, $output_i = input_i$ else $output_i = 0$ |
| Keep Odds | Returns an array of equal length to that of the input array. If $input_i$ is odd, $output_i = input_i$ else $output_i = 0$ |
| Keep Positives | Returns an array of equal length to that of the input array. If $input_i > 0$, $output_i = input_i$ else $output_i = 0$ |
| Negative | Returns the input array multiplied by -1. $output_i = -input_i$ |
| Pop from Array | Returns an array one element shorter than the input. Values are the input array's values without the last. |
| Reduce Length by Half | Returns an array half the length of the input array. Values in it are the first values of the input array. |

Table 6: Continuation of table 5

| Program | Function |
|---------|----------|
| Reverse | Returns a copy of the input array, but with the elements in reverse order |
| Shift Left | Returns an array of length one less than the input array. Values are shifted, such that $output_i = input_{(i+1)}$. The first input value is therefore not replicated in the output. |
| Shift Left Zero Padded | Returns an array of length equal to the input array. Values are shifted, such that $output_i = input_{(i+1)}$. The last value of the output array is set to zero. |
| Shift Right | Returns an array of length one greater than that of the input array. Values are the values in the input array preceded by a zero. |
| Shift Right Lossy | Returns an array of length equal to that of the input array. Values are such that $output_i = input_{(i-1)}$, with the first output value being zero. |
| Shuffle Zeros to Back | Returns an array of length equal to that of the input array. Values are the the same elements as in the input array, with the exception that all zero-values are moved to the end of the sequence. |
| Sign of | Returns an array of length equal to that of the input array. Values in output -1, 0 or 1. If $input_i > 0$ then $output_i = 1$, if $input_i < 0$ then $output_i = -1$, else $output_i = 0$ |
| Sort | Returns a copy of the input array, sorted in ascending order. |
| Square Values | Returns an array of length equal to that of the input array. Values are such that $output_i = (input_i)^2$ |
| To Iterator | Returns an array of length equal to that of the input array. Values are such that $output_i = i$ |

Table 7: Continuation of table 5. The programs in this table are those which use both the input array and the input integer.

| Program | Function |
|---------|----------|
| Add | Returns an array of length equal to that of the input array. Values are such that $output_i = input_i + inputInteger$ |
| Append | Returns an array of length one greater than that of the input array. Values are those of the input array, followed by the input integer. |
| Clip to Max | Returns an array of length equal to that of the input array. Values are such that if $input_i > inputInteger$ then $output_i = input_i$ else $output_i = inputInteger$ |
| Clip to Min | Returns an array of length equal to that of the input array. Values are such that if $input_i < inputInteger$ then $output_i = input_i$ else $output_i = inputInteger$ |
| Constant Addition | Returns an array of length equal to that of the input array. Values are such that $output_i = input_i + (i * inputInteger)$ |
| Fill Array | Returns an array of length equal to that of the input array. Values are such that $output_i = inputInteger$ |
| Greater Than | Returns an array of length equal to that of the input array. Values are such that if $input_i > inputInteger$ then $output_i = 1$ else $output_i = -1$ |
| Iterate from Start | Returns an array of length equal to that of the input array. Values are such that $output_i = i + inputInteger$ |
| Less Than | Returns an array of length equal to that of the input array. Values are such that if $input_i < inputInteger$ then $output_i = 1$ else $output_i = -1$ |
| Multiples of | Returns an array of length equal to that of the input array. Values are such that $output_i = i * inputInteger$ |
| Subtract | Returns an array of length equal to that of the input array. Values are such that $output_i = input_i - inputInteger$ |