# OpenReview forum: "Discriminator Based Corpus Generation for General Code Synthesis"
_ICLR.cc/2020/Conference — Reject_

### Official Review · AnonReviewer1 · 2019-10-18
**Official Blind Review #1**

**Rating:** 1

**Review:**

= Summary
The submitted paper describes a system for data augmentation and program synthesis given input-output examples. The system consists of (a) a discriminator able to recognise I/O examples that are likely to originate from programs humans are interested in, (b) a method to generate new example corpora using the generator from (a), and (c) a program synthesiser trained on the family of corpora generated in (b).

= Strong/Weak Points
+ The idea of interpreting the "human-likeness" of program behaviors is interesting, and could help substantially with augmenting the traditionally small clean datasets in program synthesis. It's substantially different from the competing idea of using a language model to generate "natural programs" from which examples are then extracted.
- Many important technical details are not discussed in the paper (Sect. 3.5: how are programs mutated?; Sect. 3.6: how are the predictions of networks trained on different corpora combined?)
- The writing is confusing in many places (see my pre-review clarification questions)
- The experiments leave important questions uncovered:
   (i) does the corpus-generation procedure bias the synthesizer towards human-like programs, or towards programs from the initial program corpus? This could be tested by holding out some of the programs from the initial set, and testing on these?
   (ii) random generation of programs is much cheaper than the presented strategy, but Sect. 4.2 compares based on the number of examples. How long does generating the 5 subcorpora used to generate the "collated discriminated corpus" take? How many random examples can you generate in that time? How does a system trained on that larger set perform?

= Recommendation
In its current state, I believe that the paper is lacking the technical precision and experiments to be useful to other researchers, and hence recommend rejecting it.

= Minor Comments
- Please use \citet/\citep to make sentences with citations more readable.
- Please distinguish human-useful "program behavior" and "program"

**Experience Assessment:**

I have published in this field for several years.

**Review Assessment: Checking Correctness Of Derivations And Theory:**

N/A

**Review Assessment: Checking Correctness Of Experiments:**

I assessed the sensibility of the experiments.

**Review Assessment: Thoroughness In Paper Reading:**

N/A

---

### Official Review · AnonReviewer2 · 2019-10-21
**Official Blind Review #2**

**Rating:** 1

**Review:**

This paper presents a technique based on genetic programming to generate a suitable training corpus of programs and I/O examples using a trained discriminator network. Given a set of human useful programs and the corresponding I/O examples, the main idea of the approach is to use genetic programming to iteratively construct new corpus leveraged by a discriminator that aims to make the I/O examples of the corpus closer to the I/O examples of the human-useful set. This approach is evaluated on 40 human useful corpus of array programs and it is able to synthesize more programs than random corpus or genetic programming based baselines.

Overall, this paper presents an interesting idea of automatically generating corpus for training neural program synthesis architectures, where most previous techniques sample synthetic programs uniformly from the space of DSL programs. (Shin et al. 2019) also point out a similar issue in neural program synthesis approaches, but this paper presents an automated technique to construct a better training corpus of synthetic programs and the corresponding I/O examples. I like the idea and the overall direction, but the current paper looks a bit preliminary both in evaluation as well as presentation.

First, the description of the overall method is too high-level. It would be better to formalize exactly the network architectures for both synthesis as well as the discriminator networks. Having precise inputs to the networks as well as equations would help with the description. The description of the genetic programming framework to generate child corpus from parent corpus also seems a bit high-level.

It wasn’t clear what are the inputs to the discriminator network (in Section 3.5). Does it only take a single I/O example as input or a pair of I/O examples? Does it also take the current corpus program as input? How is the discriminator network used to select programs for the child corpus?

I was trying to better understand the context in which such a dataset generation might be useful. It seems for such a technique, one has to come up with a dataset of human useful corpus of tasks. Doesn’t it mean that one has to possibly identify all possible programs that a user might want to synthesize upfront? It would be interesting to evaluate how well the technique works when synthesizing for programs that are also interesting to users but not provided as a part of the human useful corpus.

It was surprising to see genetic programming baseline solving more unique programs in the test set. Is it the case that with the discriminator based corpora, the approach is overfitting to one class of problems and not on others? Evaluating the technique on unseen programs not in the human useful set might help better evaluate this point as well.

For the comparison in 4.2 with uniformly generated data, only 5000 programs were used for training, unlike typical approaches that train on millions of synthetic programs. Why not train the network on larger amount of programs?

It was also interesting to see that the identity program was difficult to synthesize. Wouldn’t enumeration based approaches first start with that identity program? From the description it was also not clear how big the total search space for programs was in the language. Why wouldn’t enumeration based approaches work here? Also, it would be good to better understand why Sketch (Solar-Lezama 2008) based symbolic approaches won’t be able to synthesize these programs.

It would also help to provide some examples of synthesized programs and their I/O examples, and maybe also the child corpus that lead to successfully synthesizing them.

The paper mentions that DeepCoder based baseline isn’t applicable in this setting because of the generality of the language. Is the search space too large? Alternatively, can the presented technique be applied to DeepCoder to train on a generated corpus to improve its performance? Karel (Shin et al. 2019) might be another domain used in previous literature to show the usefulness of such an approach.

On page 4, the paper mentions that the first 3 examples were provided with a specific form.  Is there any intuition why such examples are useful and whether such constraints are used for all newly generated corpus as well? What happens when only random I/O examples are used?

Minor:

References are not formatted correctly in the paper.
page 4: value between 8 and 8


**Experience Assessment:**

I have published in this field for several years.

**Review Assessment: Checking Correctness Of Derivations And Theory:**

I carefully checked the derivations and theory.

**Review Assessment: Checking Correctness Of Experiments:**

I carefully checked the experiments.

**Review Assessment: Thoroughness In Paper Reading:**

I read the paper thoroughly.

---

### Official Review · AnonReviewer3 · 2019-10-24
**Official Blind Review #3**

**Rating:** 1

**Review:**

# Summary

This paper describes an approach for generating synthetic training corpora for neural program synthesis systems. It
assumes a test set of tasks to be solved, and designates a special "discriminator" network to force a distribution of
I/O examples derived from randomly sampled programs to be as close as possible to the I/O examples supplied for the
desired programs. The system first randomly samples a set of programs, then retains only those that are "sufficiently
close" to the desired I/O examples, and mutates them to further refine this training corpus. The whole approach is
evaluated on a DSL of array-manipulating programs, assuming 40 reasonable test programs and randomly sampled I/O pairs.

# Strengths

I like the core idea of the paper: assuming some desired distribution of I/O examples, construct the training corpus of
random programs in such a way that its associated I/O examples approximate the desired distribution. I can see how it
could force the sampling process to discover useful program snippets for training. If evaluated properly, this work
might get accepted at a future conference.

# Weaknesses

I have some issues with the (a) evaluation, and (b) presentation of the work in its current state.

First, the evaluation process fixes a set of 40 "useful" programs along with their associated I/O pairs, uses these I/O
pairs to build the training corpus as described, and then tests the finally trained synthesizer on the **same** set.
While the test programs themselves are not are not used in corpus building, only their I/O pairs are, this is still
sufficient to significantly bias the training set toward the test set. To properly test the core idea above, I'd expect
_at least_ something like:
- Write down or collect a set of "useful" test programs. Ideally make sure they all use different compositions of the
  language operators but still cover the space of combinatorial operator combinations well.
- Split this set into "dev" and "test" randomly. Use the I/O pairs from the "dev" set to build the training corpus.
  Evaluate the final synthesizer on the test set.
- Ideally also evaluate the sensitivity of this process to the chosen dev/test split.

Second, the evaluation baselines for corpus generation are straightforward: random sampling and genetic programming. The
former is self-evidently strawman. The latter is interesting, but is effectively a different implementation of the same
core idea of the paper, with two changes: (a) hardcoded fitness function instead of a trained discriminator, (b)
tournament selection for population evolution instead of roulette.
In any case, neither of the baselines are truly independent strong alternatives. The authors remarked that they were
"unable re-use baselines from ... literature, as for example ... DeepCoder framework caused by the relative generality
of our programming language for synthesis", which is a confusing statement for two reasons:
1. The programming language considered is a simple DSL of array manipulations, not that different in spirit from the
   DeepCoder DSL, AlgoLisp [Polosukhin & Skidanov, 2018], or the list manipulations DSL of Ellis et al. [2018].
2. Regardless of the DSL, what is being compared are **corpus generation techniques, not synthesis algorithms**.
   DeepCoder is not a baseline for that, its corpus generation is not a contribution.
The most proper baseline would be the work of Shin et al. [2019], which the authors mention in Section 2. It was
evaluated on Karel, which leaves the authors the choice of either (a) additionally evaluate their technique on the
Karel DSL, or (b) adapt the method of Shin et al. by designing some salient variables for the DSL of this paper.
I would prefer (a), which would significantly strengthen the paper, but (b) is also acceptable assuming honest effort
in designing the salient variables to make the baseline strong.

Finally, the presentation of the technical sections is informal and often confusing. Section 3 required several passes
to understand the corpus generation process. The authors should introduce proper formalism for all the involved concepts
such as corpora, train/test programs, their associated I/O pair, both neural networks with their input-output signatures
and architecture, and so on. Most of these concepts have established mathematical notation in the literature which would
make Section 3 much easier to follow. In addition, Sections 3.3-3.5 need to be formalized in algorithmic pseudocode.

# Minor remarks

* Please use \citet and \citep appropriately: citations should be enclosed in parentheticals if they don't participate
  in a sentence.
* In your synthesizer network architecture, do you know the number of program output lines ahead of time?
* In this DSL, how many possibilities are there in total for an output line (including all its arguments)?
  Given the beam size of 1M (which is a lot!), I want to compare it with the total space (#possibilities × #lines).


# References

Ellis, K., Morales, L., Sablé-Meyer, M., Solar-Lezama, A., & Tenenbaum, J. (2018). Learning libraries of subroutines for
neurally–guided bayesian program induction. In Advances in Neural Information Processing Systems (pp. 7805-7815).

Polosukhin, I., & Skidanov, A. (2018). Neural program search: Solving programming tasks from description and examples.
arXiv preprint arXiv:1802.04335.

Shin, R., Kant, N., Gupta, K., Bender, C., Trabucco, B., Singh, R., & Song, D. (2019). Synthetic Datasets for Neural
Program Synthesis. In ICLR.


**Experience Assessment:**

I have published in this field for several years.

**Review Assessment: Checking Correctness Of Derivations And Theory:**

I carefully checked the derivations and theory.

**Review Assessment: Checking Correctness Of Experiments:**

I assessed the sensibility of the experiments.

**Review Assessment: Thoroughness In Paper Reading:**

I read the paper thoroughly.

---

### Comment · AnonReviewer1 · 2019-10-09
**Clarifications**

I've tried my best to understand this paper, but I have a number of questions whose answers would be crucial for an informed review:

(1) My impression is that the concepts "program" and "program behaviour" (as exemplified by I/O examples) are used interchangeably. Consequently, I parse sentences such as "This trained discriminator therefore returns an estimate $F_d$ for how human-like a program is" as "This trained discriminator therefore returns an estimate $F_d$ for how human-like the I/O examples are". Is this correct?

(2) I don't understand what the discriminator network is operating on. The paper says "The featurisation of programs is identical to our synthesis neural network (as described above)", but then follows that by "We train this network by providing all of our I/O examples for all unfound human-useful target programs" - is the discriminator operating on programs or I/O examples?

(3) I don't understand what the "overall system" is. I understand that you generate a set of training corpora, but then things become fuzzy: Is a single synthesizer trained on all of these corpora together, or is there one synthesizer trained on each corpus? If the former, how are the individual results combined?

(4) What are you training/testing on? I understand that "human-useful problems" are used to drive the generating of training corpora, but are these the same that are used to evaluate the systems in Table 1 and Table 2?

Clarifying these points would make it much easier to write an informed review.

---

> ### Author Response · Authors · 2019-10-11
> **Reply to request for clarifications**
>
> Thank you for your time reviewing our work and your questions regarding it. In answer to your questions:
>
> 1) Yes, the discriminator receives a set of I/O featurizations of a program's behavior, and so only receives the inputs and outputs from that program, never its source code. This allows it to discriminate between human useful programs, for which no source code is ever provided, and generated programs.
>
> 2) The discriminator works on I/O examples, and is trained to learn which I/O transformations are most similar to those seen in the human-provided I/O tasks. The trained discriminator can *then* be used to guide program (actual lines of code) generation and retention in our corpus generation process. In essence, a corpus is generated via genetic programming, the descriminator is used to learn which I/O from that corpus is most similar to human useful examples, and the programs which are responsible for that most-human-like-I/O are then carried forward into a new corpus. The descriminator's estimates for human-like I/O patterns are correlated to the presence of certain programmatic features in the underlying source code, despite the discriminator never seeing this source code.
>
> 3) Our system provides both options and we experimentally evaluate both. Our primary design is that one synthesizer neural network is trained on each corpus, and we invoke each such trained network in turn when presented with an new I/O example to attempt to infer a matching program (this is explored in Sec. 4.1). It is also naturally possible to train a collated corpus, although our preliminary results with a simple network design indicates less impressive inference results than the set of individual experts (shown in Sec. 4.2).
>
> 4) In essence, contrary to typical neural network research, we do not have a training set, only a testing set. As no source code is provided, this research could be considered as a system with no externally provided training data, only a set of testing I/O examples, from which the system derives its own training corpus of source code programs. It is indeed then true to say that "human-useful problems" are used to drive the generating of training corpora, and are the same that are used to evaluate the systems in Table 1 and Table 2.
>
> If you have any further questions, don't hesitate to ask.

---

### Author Response · Authors · 2019-11-12
**Response to Reviewers**

Dear reviewers,

We recognise and acknowledge your reviews, and at this point are not asking for you to change your scores. However, as the reviews are open to general viewing, we are submitting a response to answer the main points raised.

A core point of ambiguity on our part is the intended focus of the paper. We intended this system primarily as a training corpus generation tool, and presented the overall results as a means to evaluate the performance of the generator, with the neural network itself being deliberately simple (although not deliberately underperforming). As such, we would respectfully argue that it is not a strawman to compare against random sampling, as this technique is a very commonly used approach to corpus generation, in this field and others. Random sampling would presumably be the training-corpus generation method of choice for any researcher in the field unable to access a human-generated training corpus and working in a problem space as large as ours.

We purposefully chose a very large search space, as existing techniques work very well on currently studied domains (Karel and DeepCoder's DSL). These problem spaces are tractable to existing training set generation techniques, but we strove to demonstrate that these techniques do not provide useful training corpora on larger problem spaces, which are inevitable with increases in language expressiveness. In response to the suggestion to provide information about contextualising our 1,000,000 searches (which the reviewer implied was a substantial portion of the search space), there are 2195 options per line, therefore giving a search space size of around 6*10^46, with preliminary experiments giving an estimated lower bound of 10^22 unique functionalities. It was in response to this daunting search space that we found random sampling of program space to be an insufficient tool, and one in which a search of 10^6 programs could only find a valid solution if the network's output were near-perfectly accurate.

Regardless of these disagreements, allow us to thank the reviewers for the constructive feedback, and for the support of the underlying ideas. We very much hope to turn your reviews into better science.

Yours,
Anonymous authors 1 & 2

---

### Decision · Program_Chairs · 2019-12-19

**Decision:**

Reject

**Comment:**

This paper proposes a method to automatically generate corpora for training program synthesis systems.

The reviewers did seem to appreciate the core idea of the paper, but pointed out a number of problems with experimental design that preclude the publication of the paper at this time. The reviewers gave a number of good comments, so I hope that the authors can improve the paper for publication at a different venue in the future.